# Prediction of DNA Methylation based on Multi-dimensional feature encoding and double convolutional fully connected convolutional neural network

**Wenxing Hu, Lixin Guan, Mengshan Li**⬵*

College of Physics and Electronic Information, Gannan Normal University, Ganzhou, Jiangxi, China

* msli@gnnu.edu.cn

## Abstract

DNA methylation takes on critical significance to the regulation of gene expression by affecting the stability of DNA and changing the structure of chromosomes. DNA methylation modification sites should be identified, which lays a solid basis for gaining more insights into their biological functions. Existing machine learning-based methods of predicting DNA methylation have not fully exploited the hidden multidimensional information in DNA gene sequences, such that the prediction accuracy of models is significantly limited. Besides, most models have been built in terms of a single methylation type. To address the above-mentioned issues, a deep learning-based method was proposed in this study for DNA methylation site prediction, termed the MEDCNN model. The MEDCNN model is capable of extracting feature information from gene sequences in three dimensions (i.e., positional information, biological information, and chemical information). Moreover, the proposed method employs a convolutional neural network model with double convolutional layers and double fully connected layers while iteratively updating the gradient descent algorithm using the cross-entropy loss function to increase the prediction accuracy of the model. Besides, the MEDCNN model can predict different types of DNA methylation sites. As indicated by the experimental results, the deep learning method based on coding from multiple dimensions outperformed single coding methods, and the MEDCNN model was highly applicable and outperformed existing models in predicting DNA methylation between different species. As revealed by the above-described findings, the MEDCNN model can be effective in predicting DNA methylation sites.

## Author summary

DNA methylation is an important DNA modification form associated with a wide range of biological processes.Identifying accurately methylation sites on a genomic scale is crucial for under-standing of biological functions. This study proposes an algorithm based on Multi-dimensional feature encoding and double convolutional fully connected

**Data Availability Statement:** The codes, architecture, parameters, dataset, functions, usage and output of the proposed model are available free

**Funding:** The Natural Science Foundation of China supported financially this work: 51663001, 52063002 and 42061067 to ML. The funders had no role in study design, data collection and analysis, decision to publish, or preparation of the manuscript.

**Competing interests:** The authors have declared that no competing interests exist.

convolutional neural network to predict different types of DNA methylation sites. As indicated by the experimental results,the deep learning method based on coding from multiple dimensions outperformed single coding methods, and the MEDCNN model was highly applicable and outperformed existing models in predicting DNA methylation between different species.The results showed that our method could accurately predict the DNA methylation sites in different species.

## Introduction

DNA methylation refers to a genetic expression modification [1] that has been extensively investigated; it takes on critical significance in cell growth, differentiation, and other life processes [2–5], as well as in the regulation of gene expression. To be specific, It is a type of DNA chemical modification, through which a methyl group, provided by S-adenosylmethionine (SAM), can be covalently bonded to the cytosine 5 carbon position in the DNA strand, such that 5-methylcytosine [6] (5mC) can be formed under the catalysis of DNA methyltransferase (DNMT). Besides 5mC, there are other types of methylation, comprising N6-methyladenine [7–9] (6mA) and N4-methylcytosine [8] (4mC), with their chemical structures illustrated in Fig 1. The identification of DNA methylation modification sites takes on vital significance essential as a key genetic manifestation that can be conducive to gaining insights into the mechanisms of gene regulation(e.g., the induction of abnormal proliferation leading to cancer) [10–14].

Whole-genome bisulfite sequencing (WGBS) has been employed as the conventional method to detect DNA methylation, and treatment of DNA with bisulfite can convert cytosine residues (C) to uracil (U), whereas 5-methylcytosine residues (5mC) are resistant to it and not subjected to conversion [15]. Accordingly, DNA administrated with bisulfite retains only the methylated cytosine [16], which is subsequently combined with high-throughput sequencing technology [17] and compared with a reference sequence [18]. However, this transformation leaves less 'C' and more 'A', 'G', and 'T' in the genome, which is not the case in biology, while the general reference genome is employed in the comparison. Besides, all the above-mentioned transformed sites cannot be matched to the corresponding loci in the reference genome. The detection of DNA methylation using the above conventional method is time-consuming and labor-intensive. The existing research direction has been more concerned with the

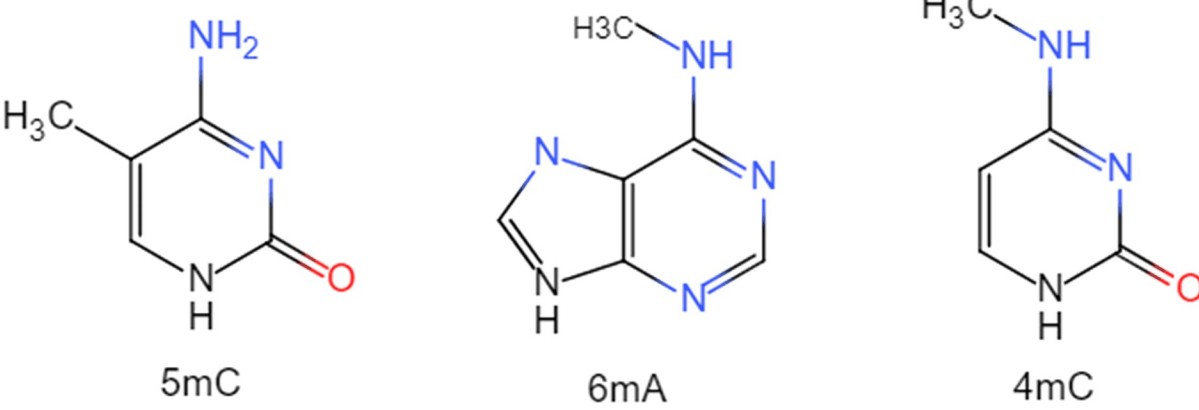

**Fig 1. Chemical structure of 5mC,6mA,4mC.**

development of computational methods [19]. Several machine learning methods have been explored and employed to predict DNA methylation modification sites.

With the prediction of 6mA methylation type as an example, the SNNRice6mA [20] model employs a convolutional neural network (CNN) to identify 6mA sites in the rice genome. Subsequently, Li et al. proposed Deep6mA [21], i.e., a hybrid deep learning model combining convolutional neural networks and long and short-term memory, which is capable of predicting 6mA modifier sites more accurately than SNNRice6mA. Similar to Deep6mA, BERT6mA [22] adopts Transformer to build the model. This model does not differ significantly from Deep6mA in prediction results while demonstrating the application of natural language processing techniques in predicting 6mA modification sites; it achieves effective results. The DeepTorrent [23] predictor based on deep learning is effective in predicting 4mC methylation. This model combines the initial module, attention module, and migration learning to enhance the prediction performance of 4mC sites. It is noteworthy that Deep4mC [24] can extend the deep learning framework using bootstrap methods to more effectively predict 4mC loci for species with small sample sizes. However, most of the methods are limited to predicting a single type of DNA methylation modification sites [20–36], and these methods are difficult to use in other types. The iDNA-MS [37] model first extracts features using three coding sequence methods and then predicts three types of DNA methylation modification sites using random forests. However, it employs a conventional machine learning method, and the performance of this model has considerable room for improvement. The latter has been designed to enhance the performance of prediction by iDNA-AB [38] and iDNA-ABT [38]. Both of them use a bidirectional encoder BERT representation of the architecture to automatically learn distinguishable features and then make predictions for a wide variety of methylation sites. iDNA-ABT uses TIM loss, which is not consistent with iDNA-AB where the loss function employed in the classification module is cross-entropy. Both of them make relatively accurate predictions for different types of methylation sites. As mentioned above, there has been an increasing number of studies exploring deep learning applications for predicting DNA methylation, and their prediction performance has been significantly improved. However, existing deep learning methods of predicting DNA methylation have not sufficiently explored the features of DNA gene sequences for learning to uncover the vital effect of gene sequences in predicting DNA methylation.

Although machine learning-based methods fulfill the objective of predicting DNA methylation modification sites, they differ significantly in several details (e.g., the encoding of the sequence features applied and the model structure). A method that investigates the structure of a fusion model with gene feature information in depth can determine its prediction performance. Thus, enhancing the performance of the model is critical to the research of novel methods. In this study, a deep learning-based method for DNA methylation modification site prediction was proposed, termed the MEDCNN model. In MEDCNN, the sequence was encoded and fused using positional, chemical, and biological information of the DNA gene sequence. To build a robust model, the CNN parameters of the MEDCNN model were tuned, the combination of parameters with the optimal prediction performance was selected for training, and the cross-entropy loss function was optimized iteratively using Gradient descent, such that different types of DNA methylation modification sites can be more effectively predicted.

## Materials and methods

### Datasets

The benchmark dataset provided by the universal DNA methylation predictor-iDNA-MS [37] was adopted in this study, containing a total of 17 datasets encompassing methylation

modification sites of different species in three major types (i.e., 4mC, 5hmC, and 6mA). To avoid redundancy and reduce homology bias, Lv et al. (37) used the CD-HIT procedure to remove sequences with more than 80% sequence similarity. Among the 4mC methylation types, the dataset covered four species of *Casuarina equisetifolia* (C.equisetifolia), *Fragaria vesca* (F.vesca), *Saccharomyces cerevisiae* (S.cerevisiae), as well as *Tolypocladium sp SUP5-1* (Tolypocladium). Among the 6mA methylation types, the species covered by the dataset involved *Drosophila melanogaster* (D.melanogaster), *Caenorhabditis elegans* (C.elegans), *Arabidopsis thaliana* (A.thaliana), *Rosa chinensis* (R.chinensis), *Tetrahymena thermophila* (T. thermophile), *Xanthomonas oryzae pv. Oryzicola (Xoc) BLS256* (Xoc.BLS256), *Homo sapiens* (H. sapiens), D.melanogaster, F.vesca, S.cerevisiae, as well as Tolypocladium. Among the 5mC methylation types, the dataset comprises two species (i.e., M. musculus and H. sapiens). Due to the dataset provided by Lv et al. (37), where the ratio of training set to test set is 1:1, it is not suitable for the data partitioning commonly used in machine learning. Therefore, we combined the sequences from the training and test sets of the original data set for each species separately. Subsequently, we redistributed the data into new training and test sets, following a ratio of 7:3 or 8:2. The sequences in the training and test sets exist independently. Furthermore, within the training set, 10% of the data was set aside as a validation set. During training, the validation set was adopted to examine the generalization ability of the model and the presence of overfitting. Besides, after training, the performance of the network was evaluated through the test. The model was trained on the datasets of various species for the methylation types 4mC, 5hmC, and 6mA. This training was conducted in order to predict the corresponding methylation sites for these species. Table 1 lists the details of the 17 datasets.

## Feature encoding of multiple dimensions

Three different ways of encoding DNA gene sequence features based on four DNA base sequence types (i.e., 'A', 'G', 'C', and 'T') were employed. Moreover, the gene sequences close to a segment of DNA methylation modification sites were converted into a digital feature matrix. To facilitate the description of sequence features, the DNA sequence can be denoted as $S = D_1 D_2 \ldots D_i$, where $D_i \in (A,G,C,T)$ represents the deoxyribonucleotide at the i-th position in the sequence. Furthermore, the above-described three encoding methods were classified into three dimensions (i.e., location-based, physicochemical property-based, and biological

**Table 1. Number of gene sequences in the training and test sets.**

| Dataset | 5hmC | | 4mC | | 6mA Heading4 | |
|---|---|---|---|---|---|---|
| | Training data | Testing data | Training data | Testing data | Training data | Testing data |
| H.sapiens | 3577 | 584 | - | - | 29312 | 7356 |
| M.musculus | 5947 | 1409 | - | - | - | - |
| C.equisetifolia | - | - | 3165 | 791 | 9492 | 2638 |
| F.vesca | - | - | 27385 | 4207 | 4928 | 1274 |
| S.cerevisiae | - | - | 3272 | 684 | 6053 | 1517 |
| Tolypocladium | - | - | 21400 | 9250 | 5295 | 1461 |
| D.melanogaster | - | - | - | - | 15668 | 6712 |
| R.chinensis | - | - | - | - | 895 | 301 |
| Xoc BLS256 | - | - | - | - | 27460 | 6968 |
| C.elegans | - | - | - | - | 12741 | 3179 |
| T.thermophile | - | - | - | - | 11243 | 4677 |
| A.thaliana | - | - | - | - | 45282 | 18462 |

property-based). On that basis, feature information was extracted to fuse to assist deep learning models in predicting DNA methylation modification sites.

## Binary encoding of Position Feature (BPF)

BPF is equivalent to One-hot coding, i.e., a sparse binary, 4D word vector [37,39] hat provides position-specific nucleic acid information. It simply encodes the DNA sequence as a feature matrix based on the position-specific structure of the DNA nucleic acid sequence, where each nucleic acid is represented by a 4D binary vector (0/1) [40–42]. The calculation is shown in Eq 1).

$$b = \begin{cases} (1,0,0,0), if \ b = A \\ (0,1,0,0), if \ b = T \\ (0,0,1,0), if \ b = G \\ (0,0,0,1), if \ b = C \end{cases} \tag{1}$$

Thus, for a given segment of DNA gene sequence of length L, it can be converted into a $4 \times L$ feature matrix.

$$A_1 = \begin{bmatrix} BPF_1(1) & \cdots & BPF_1(L) \\ \vdots & \ddots & \vdots \\ BPF_4(1) & \cdots & BPF_4(L) \end{bmatrix} \tag{2}$$

## Coding of nucleic acid chemical properties (NCP)

The four deoxyribonucleotides of DNA cover different bases, and their differences are specific to hydrogen bond strength, ring structure and biological function [43,44]. For the differences in ring structures, 'A' and 'G' cover two rings, whereas 'T' and 'C' have only one ring. For the hydrogen bond strength, 'G' and 'C' form strong hydrogen bonds between each other, while 'T' and 'A' form a weak hydrogen bond. For the other components, 'T' and 'G' belong to the ketone group, and 'A' and 'C' belong to the amino group. Thus, according to the above-mentioned three classifications, the coding of DNA gene sequences can be classified as shown in Eq 3 (where $i$ denotes the position of the base in the DNA sequence).

$$NCP_1(i) = \begin{cases} 1 \ if \ D_i \in \{G, A\} \\ 0 \ if \ D_i \in \{C, T\} \end{cases}, \ NCP_2(i) = \begin{cases} 1 \ if \ D_i \in \{T, A\} \\ 0 \ if \ D_i \in \{C, G\} \end{cases}, \ NCP_3(i) = \begin{cases} 1 \ if \ D_i \in \{A, C\} \\ 0 \ if \ D_i \in \{G, T\} \end{cases} \tag{3}$$

According to the above-described three ways, 'A', 'G', 'C', 'T' can be encoded as (1, 1, 1), (1, 0, 0), (0, 0, 1), (0, 1, 0)), respectively. Thus, a DNA sequence of length L can be transformed into a $3 \times L$ feature matrix using NCP [37,39].

$$A_2 = \begin{bmatrix} NCP_1(1) & \cdots & NCP_1(L) \\ \vdots & \ddots & \vdots \\ NCP_3(1) & \cdots & NCP_3(L) \end{bmatrix} \tag{4}$$

## Coding of Dinucleotide physical and chemical properties (DPCP)

Consecutive combinations of DNA bases exhibit different physicochemical properties, i.e., a vital feature for genome structure prediction. Goni et al [45]. performed statistical predictions of the physical and chemical structural features of gene sequences based on gene structure and

homology conservation features. As revealed by their results, there is a hidden set of coding schemes in regulating genome expression: DPCP [39]. This physical coding set comprises three angular parameters (i.e., Twist, Tilt, and Roll) and three distance parameters (i.e., Shift, Slide, and Rise) in the spatial structure. To be specific, Tilt, Roll, and Twist can indicate the angular variation of the spatial plane of adjacent bases up and down, back, and forth, and left and right, respectively; Rise, Slide, and Shift can indicate the changes in distance between adjacent bases in the up and down, front, and back, and left and right relative positions, respectively [45]. The above-mentioned base duplex structure information values were obtained from previous work [46], and since the above-described six values vary in different ranges, a normalization method was adopted to scale them to the range [0,1] as expressed in Eq 5. Thus, DPCP is capable of converting a gene sequence of length L into a $6 \times$ (L-1) feature matrix.

$$X_{nom} = \frac{X - X_{min}}{X_{max} - X_{min}}$$

(5)

To ensure that the number of columns of the matrix is the same as that of the other coding schemes, the sliding dimer window algorithm was adopted to calculate the DPCP value of the respective combined base as in Eq 6 ($DPCP_n(i)$ represents the ith physicochemical property of the nth base in the gene sequence; $X_n$ expresses the nth nucleotide physicochemical property).

$$DPCP_n(i) = \frac{X_n(D_{i-1}D_i) + X_n(D_iD_{i-1})}{2}$$

(6)

Notably, the values at the ends of the calculated matrix are dependent only on the values at the ends of the DPCP, such that a $6 \times$ L feature matrix can be obtained, describing the physicochemical properties of the gene sequence.

$$A_3 = \begin{bmatrix} DPCP_1(1) & \cdots & DPCP_1(L) \\ \vdots & \ddots & \vdots \\ DPCP_6(1) & \cdots & DPCP_6(L) \end{bmatrix}$$

(7)

## Multidimensional feature coding fusion convolutional neural network

For most DNA gene sequence data processing, a recurrent neural network architecture has been used (e.g., LSTM [30,32,47] and GRU [27]). However, as a result of the encoding scheme mentioned above, we obtain a multidimensional feature matrix. It is important to note that DNA sequence methylation sites are only correlated with the information within their very small window. Therefore, it is crucial to focus primarily on the information in close proximity to the methylation sites. Thus, in this study, a convolutional neural network was adopted to process the encoded feature matrix and build a convolutional neural network based on multidimensional feature encoding with dual convolutional layers and dual fully connected layers, abbreviated as MEDCNN. The convolutional layer extracted features not affected by the coding space transformation, and the fully connected layer processed the information extracted from the upstream convolutional layer nonlinearly. Lastly, the predicted labels were obtained after the fully connected layer and proper activation. In this study, pytorch [48] of the python package was adopted to build the model. Fig 2 illustrates the workflow of MEDCNN.

The difference between MEDCNN and previous CNN [49–52] is elucidated as follows. The input to MEDCNN comprises multiple dimensions, including location information, physicochemical information, and biological information. To effectively integrate these multidimensional features, we incorporated a convolutional block attention module into the MEDCNN extraction process. By multiplying the input feature maps with the channel weights and spatial

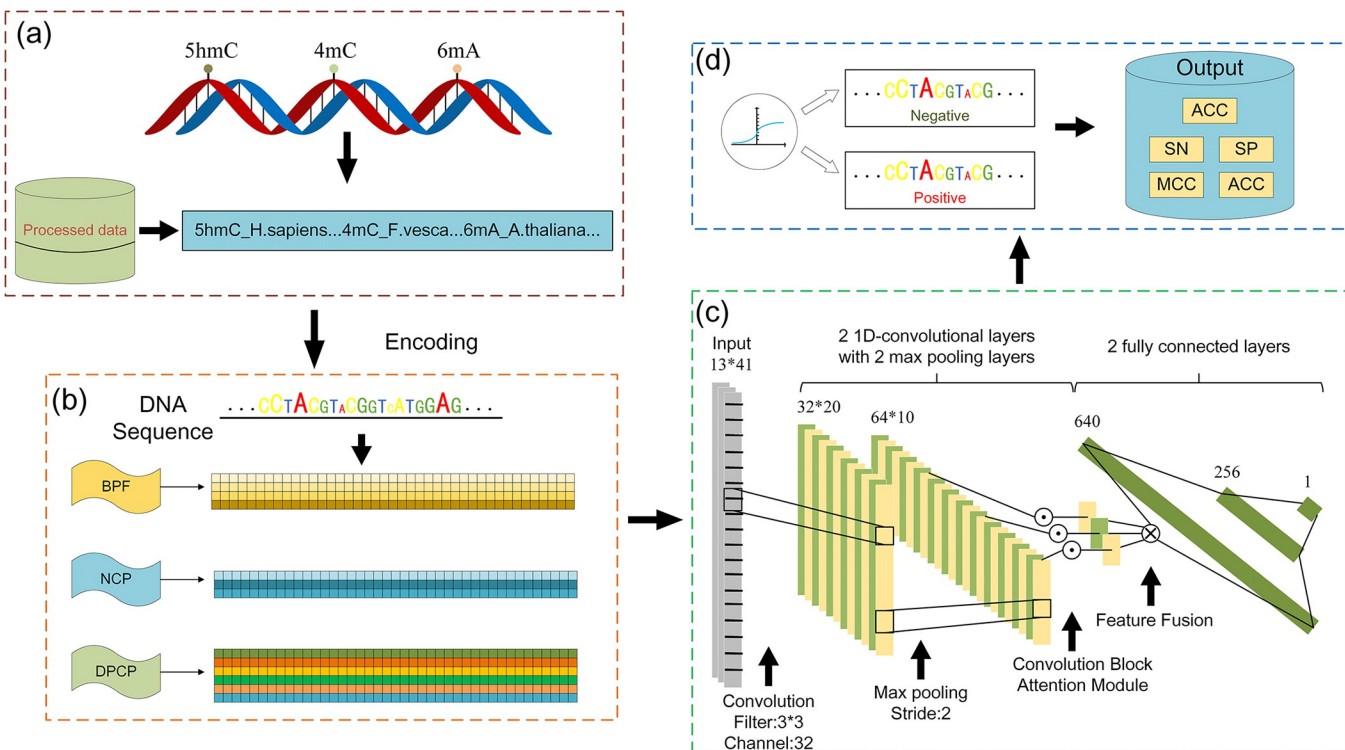

**Fig 2. The workflow of the proposed MEDCNN model for the prediction of DNA methylation sites. (a)** Dataset collecting **(b)** Feature encoding **(c)** Predictive model construction **(d)** Model performance evaluation.

weights generated by the attention module, MEDCNN gains the ability to discern significant features and their respective locations across multiple channels and spatial axes. To provide a more illustrative representation, we visualize the attention maps of the multidimensional matrix of MEDCNN inputs, as well as the multidimensional features extracted both before and after employing the attention module. Fig 3 presents these visualizations, where distinct colors denote varying weights. The hierarchical multidimensional

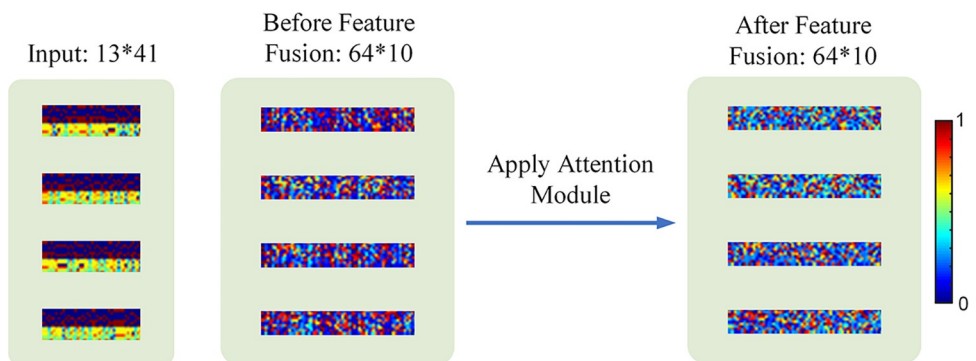

**Fig 3. Visualization of the multidimensional matrix of MEDCNN inputs and the attention graph of multidimensional features extracted before and after applying the attention module.**

information, namely $Z_1$, $Z_2$ and $Z_3$, exists in distinct feature spaces, each representing the meaning of the corresponding dimension. However, after feature extraction in MEDCNN, distinguishing location features across different channels becomes less apparent. To address this challenge, the fusion layer within the attention module combines the multidimensional information $Z_1$, $Z_2$ and $Z_3$ through a tensor product operation. This fusion process adjusts the significance of each feature channel, allowing MEDCNN to prioritize the information associated with methylation loci exhibiting higher weight values. The attention module performs the fusion of multidimensional information $Z_1$, $Z_2$ and $Z_3$ through the following tensor product procedure:

$$Z = \begin{bmatrix} Z_1 \\ 1 \end{bmatrix} \otimes \begin{bmatrix} Z_2 \\ 1 \end{bmatrix} \otimes \begin{bmatrix} Z_3 \\ 1 \end{bmatrix} \tag{8}$$

where Z denotes the fusion tensor; $\otimes$ represents the outer product between the tensors; the constant 1 preserves the original extracted features. On that basis, $Z$ can be considered the 3D cube of all possible combinations of the three tensor spaces.

In fact, the above multidimensional feature encoding fused convolutional neural network is designed to find a mapping as follows:

$$\hat{y} = \arg max f(BPF_n(i), NCP_n(i), DPCP_n(i); W). \tag{9}$$

where $\hat{y}$ denotes the methylation predicted by the multidimensional neural network; $BPF_n$ represents the feature matrix of DNA gene sequence after BPF encoding; $NCP_n$ expresses the feature matrix of DNA gene sequence after NCP encoding; $DPCP_n$ is the feature matrix of DNA gene sequence after DPCP encoding; $W$ is the parameter of the multidimensional neural network; $f$ denotes the mapping sought by the neural network.

To find such a mapping, a loss function should be defined to measure the difference between the predicted labels and the true labels, and iteratively updated by gradient descent to minimize the loss function, making the values predicted by the multidimensional neural network more accurate. Besides, the loss function employed in this study is the cross-entropy loss function commonly used to address multiclassification problems [53]:

$$L = -\frac{1}{N}\sum_{n=1}^{N}(y^{(n)} log\, p^{(n)} + (1 - y^{(n)})log(1 - p^{(n)})) \tag{10}$$

where $N$ denotes the sample size; $y^{(n)}$ represents the binary variable; $p^{(n)}$ expresses the probability that the neural network predicts the nth sample methylation.

## Model performance evaluation

To evaluate the classification performance of the model, several commonly used classification performance evaluation metrics are used here to assess the predictive performance of the model in the same way as Lv et al [37]. The above-mentioned include Sensitivity (SN), Specifcity (SP), Accuracy (ACC), Matthews' correlation coefcient (MCC) [54,55], and Area under the working characteristic curve (AUC). The specific calculation procedure is shown below.

$$
\begin{cases}
SN = \dfrac{TP}{TP + FN} \times 100\% \\[2mm]
SP = \dfrac{TN}{TN + FP} \times 100\% \\[2mm]
ACC = \dfrac{TP + TN}{TP + FN + TN + FP} \times 100\% \\[2mm]
MCC = \dfrac{TP \times TN - FP \times FN}{\sqrt{(TP + FP) \times (TP + FN) \times (TN + FP) \times (TN + FN)}} \\[2mm]
AUC = \dfrac{\sum_{i \in pos} rank_i - \dfrac{num_{pos}(num_{pos} + 1)}{2}}{num_{pos} num_{neg}}
\end{cases}
\tag{11}
$$

where TP, TN, FP, and FN represent the number of samples with true positive, true negative, false positive, and false negative prediction results, respectively. The AUC [56] was defined as the area enclosed with the coordinate axis under the ROC [57] curve, and the value of this area was not greater than 1. Since the ROC curve was generally above the line y = x, the AUC value ranged from 0.5 to 1. The value of AUC ranges between 0.5 and 1. The performance of the model was enhanced with the the value of AUC closer to 1.0.

## Results and discussion

### Experimental results of different DNA methylation types

To evaluate the performance of the proposed DNA methylation prediction method, 17 benchmark independent datasets of three different DNA methylation types were employed to test the proposed model. The resulting prediction results are presented in Fig 4, and the mean values of the result statistics are listed in Table 2. Moreover, the corresponding data are listed in S1 Table.

As depicted in Fig 4B and Table 2, for the three DNA methylation types 5hmC, 4mC and 6mA, the mean values of ACC of accuracy were 95.39%, 76.74% and 86.61%, respectively, and the overall prediction results were good. Notably, for the 5hmC methylation type Fig 4F, the prediction accuracies of H. sapiens and M. musculus were 93.7% and 97.1%, respectively. Besides, the results of other evaluation indexes were robust, basically exceeding 70%, with an overall distribution of more than 80%. To be specific, the prediction results of 5hmC methylation type were the optimal (Fig 4D and 4E), and the evaluation metrics of predicting 5hmC, AUC and MCC, were roughly up to 90%. As revealed by the above result, the proposed multi-dimensional information extraction feature-assisted deep learning to predict DNA methylation is stable and reliable.

### Experimental results of different feature encoding

In the present section, the experiment was performed based on the following question, i.e., whether the fusion of multidimensional feature coding methods is more effective than individual coding methods in identifying DNA methylation types (5hmc/4mC/6mA) for the respective species. For the above purpose, we progressively used BPF, NCP and DPCP with their three combined coding Multi-Fe to identify the methylation sites of 17 datasets. S2 Table and Fig 5 present the experimental results achieved in this study.

In order to further investigate whether there are differences between different encoding methods in the context of machine learning, we conducted a non-parametric Wilcoxon

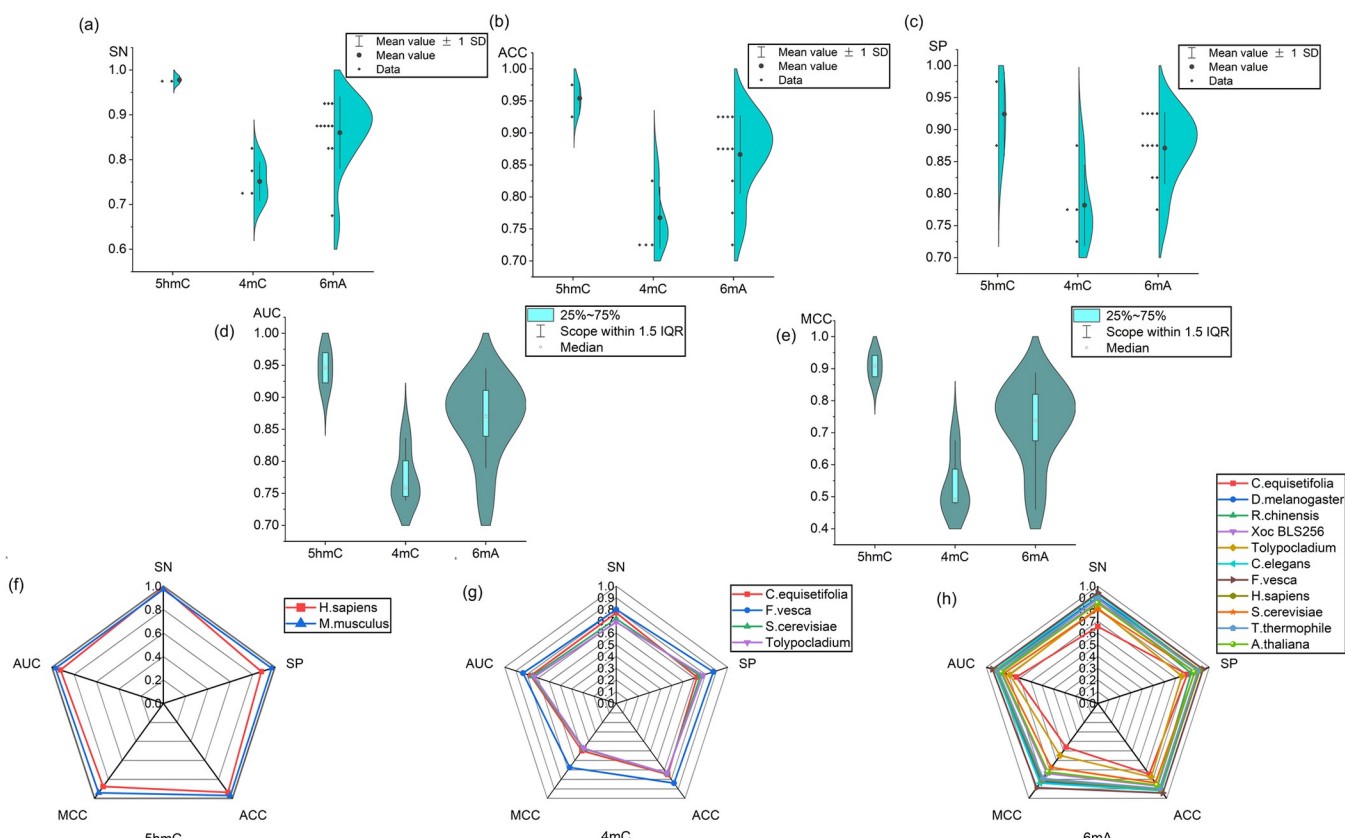

**Fig 4. The results of MEDCNN prediction of independent datasets. (a)**, **(b)**, **(c)**, **(d)** and **(e)** represent the predicted values of SN, SP, ACC, MCC and AUC for the three methylation types and the distribution of the results, respectively.The size of its contour represents the degree of concentration or clustering of the results. **(f)**, **(g)** and **(h)** illustrate the prediction indexes for identifying methylation types of 5hmC, 4mC and 6mA by using independent datasets.

signed-rank test to compare the significant differences in ACC values among them. We calculated the rank sum statistic and the corresponding p-value to assess the differences between the samples. The significance level was set at α = 0.05. When the p-value is less than α, it indicates a significant difference between the samples, and the smaller the p-value, the greater the difference. The specific results are listed in Table 3.

As depicted in Fig 5, for all three methylation types, the performance of using Multi-Fe to extract features for methylation modification site prediction was basically better than that of other coding methods. Besides, among the 4mC methylation types, the predicted ACC values of DPCP coding (Fig 5B) for C.equisetifolia and S.cerevisiae species were notably inferior to those of BPF and NCP coding, whereas the fusion of the three coding methods exhibited better performance. This demonstrates the enhancement achieved by the fusion of multidimensional

**Table 2. Predicting the mean value of different DNA methylation types.**

| Dataset | SN | SP | ACC | MCC | AUC |
|---|---|---|---|---|---|
| 5hmC | 0.9780 | 0.9242 | 0.9539 | 0.9083 | 0.9461 |
| 4mC<br>4mC_F.vesca | 0.7511 | 0.7817 | 0.7674 | 0.5340 | 0.7731 |
| 6mA | 0.8600 | 0.8709 | 0.8661 | 0.7296 | 0.8666 |
| Average | 0.8630 | 0.8589 | 0.8625 | 0.7240 | 0.8619 |

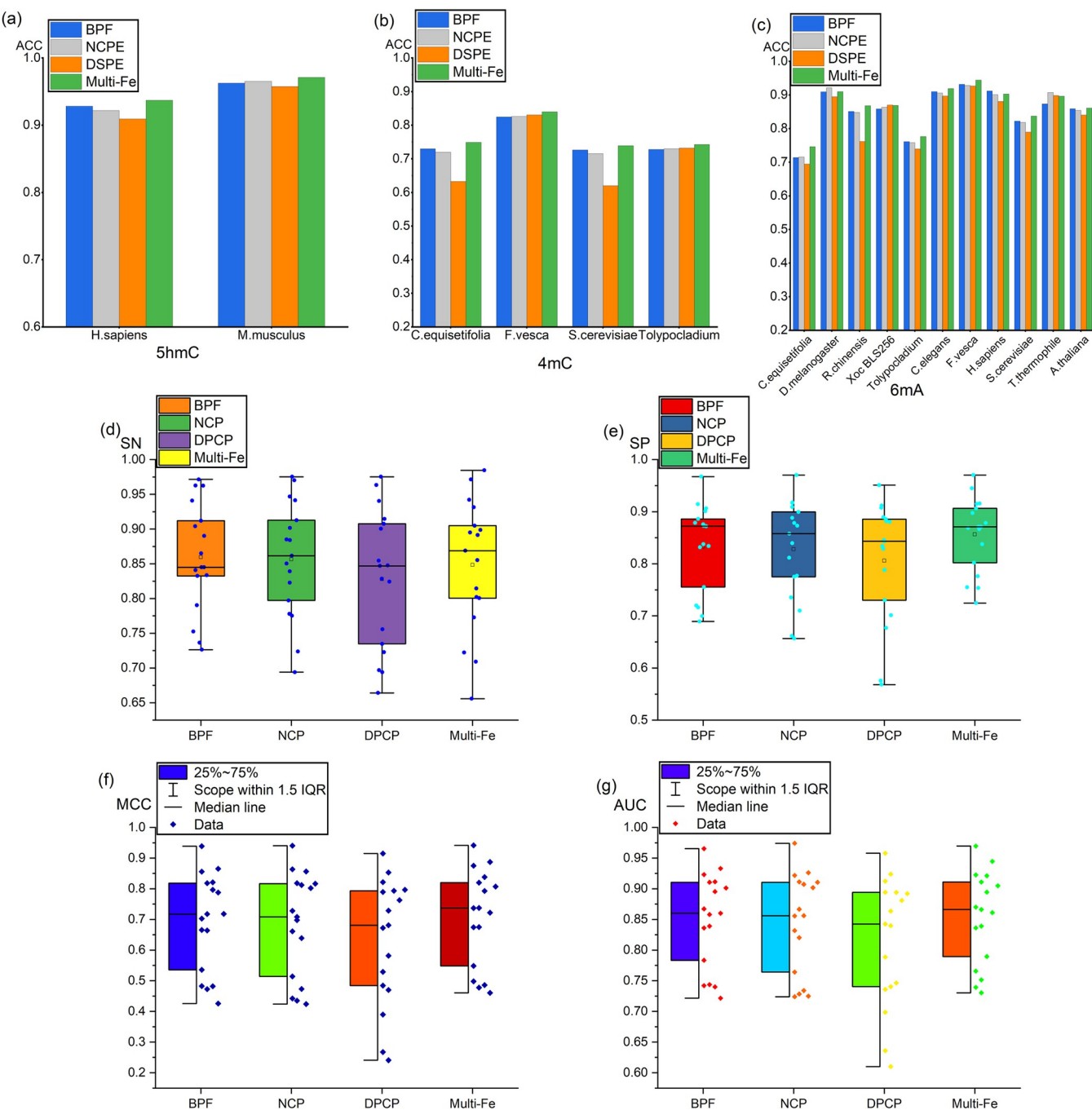

**Fig 5. Results of identifying different feature codes for 5hmc/4mC/6mA methylation types in 17 datasets.** (**a**), (**b**) and (**c**) represent the comparison of ACC values of predicted methylation types of 5hmC,4mC and 6mA with different coding methods, respectively. (**d**), (**e**), (**f**) and (**g**) represent the SN, SP, ACC and AUC values of 17 datasets with different encoding methods combined with CNN to identify 5hmC/6mA/4mC sites, respectively.

coding methods for predicting DNA methylation modification sites in different species compared with individual feature extraction, and thus multidimensional feature extraction proved to be effective in the DNA methylation prediction task. Among the predicted 6mA methylation types, the ACC values of Multi-Fe were better than the other three coding modalities for seven species. As revealed by the comparison of other evaluation metrics, the overall

**Table 3. The results of the Wilcoxon test for each feature code.**

| Sample group | ACC | |
| --- | --- | --- |
| | Wilcoxon signed-rank statistic | p-value |
| Multi-Fe: BPF | 5 | 0.000153 |
| Multi-Fe: NCP | 11 | 0.000839 |
| Multi-Fe: DPCP | 3 | 0.000076 |
| BPF: NCP | 54 | 0.306046 |
| BPF: DPCP | 22 | 0.007904 |
| NCP: DPCP | 9 | 0.000504 |

distribution of results in predicting the methylation sites of 17 species with the evaluation metrics MCC values and AUC values as shown in Fig 5F and 5G were higher than those predicted by other coding methods. The results of the Wilcoxon test are presented in Table 3. The Wilcoxon signed-rank statistic for each group of samples is smaller than the product of their sample sizes. Therefore, we only need to compare the p-values with α to determine the significance. The results indicate that for the Wilcoxon test between Multi-Fe and the other three encoding methods, the p-values are less than 0.05, suggesting that the combined feature encoding of Multi-Fe significantly outperforms individual encodings. Comparing the individual feature encoding methods, we can observe that the p-values for BPF and NCP are 0.306046, which is greater than 0.05. This implies that there is no significant difference between these two encoding methods. However, both BPF and NCP encoding methods show significant differences when compared to the DPCP encoding method. From the p-value obtained through the comparison between the combination of these three codes and the DPCP codes, as well as the p-value obtained through the comparison between individual coding methods and the DPCP codes, we can observe that the p-value decreases significantly, becoming much smaller than 0.05. This indicates that the combined codes amplify the difference between them. Based on these results, we can conclude that extracting features from multiple dimensions is an effective approach for exploring gene sequences and uncovering essential information for predicting DNA methylation.In general, the accuracy of prediction results can be increased by fusing deep learning with features of gene sequences using multidimensional information extraction for DNA methylation prediction.

## Experimental results of cross-species validation

To investigate whether the model is still reliable when multidimensional feature extraction information is adopted to predict DNA methylation modification sites in different species of the same methylation type. For this purpose, we performed a cross-species validation in the same way as the study by Lv et al [37]. Five datasets were first randomly selected in 6mA methylation types, C.elegans, C.equisetifolia, F.vesca, R.chinensis and Tolypocladium. The model was trained on DNA gene sequences of one species and then predicted on DNA gene sequences of another species, so as to predict whether the gene sequence is methylated or not. The results thus obtained are shown in Fig 6.

As depicted in Fig 6C, when training the model with the DNA gene sequences of 5 species of 6mA to predict the DNA sequences of Tolypocladium, the ACC values differed between 7.4% and 1.7%, while the results of predicting the DNA methylation modification sites of other species were slightly deviated to a certain extent, and all of them were not as good as the original species to train the model. This indicates that there are differences in the 6mA modification patterns of the above-described species. Fig 6D also indicates that the AUC values of the gene

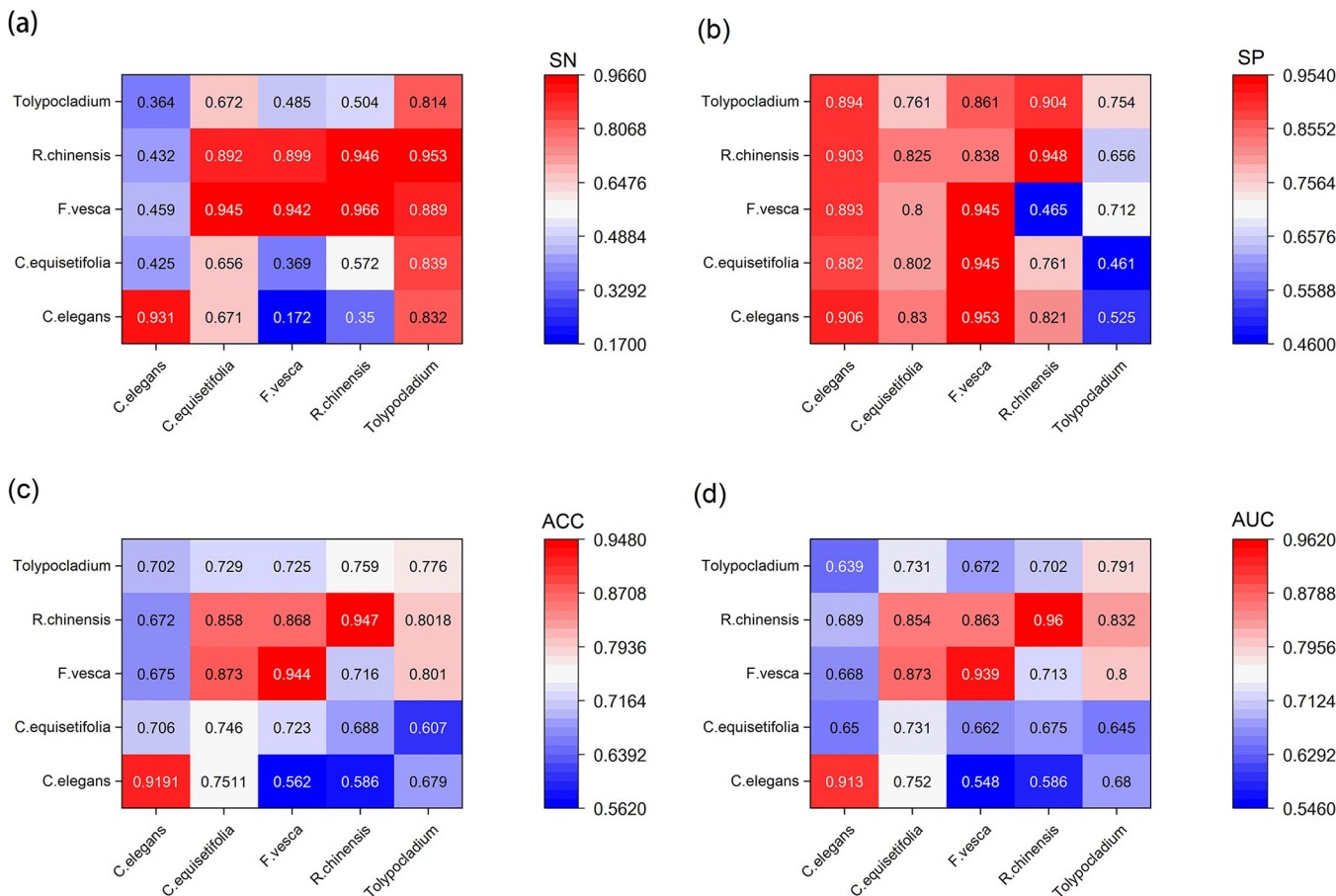

**Fig 6. Cross-species validation of 5 species of 6mA types.** The heatmaps **(a)**, **(b)**, **(c)** and **(d)** show the cross-species predicted SN, SP, ACC, and AUC values for the five species for which the 6mA methylation type was determined. Once a species has built a model on its training dataset, it was tested on data from other species. The horizontal coordinates are the different species as the training set and the vertical coordinates are as the testing set.

sequences of the four species other than the original species were primarily not as high as those of the original species after the model was trained separately, whereas some predictions were better than the others. For instance, when trained with the gene sequences of C.equiseti-folia to predict the DNA methylation sites of F. vesca modification sites, the model achieved an AUC value of 87.3%, 15.2% higher than the result of its training. In general, although the optimal accuracy was constantly obtained through prediction from models built on their data, the predictions from models built on other species took on certain significance as well. Even for gene sequences from different species, the MEDCNN model can effectively extract and uncover similar information, leading to promising prediction performance. In brief, high accuracy, robustness and applicability with strong generalization ability were reported when fusing deep learning to identify methylation modification sites by extracting feature information from DNA sequences in multiple dimensions.

## Experimental results compared with other models

To verify the feasibility of the proposed method in depth, the MEDCNN model was compared with the existing predictors iDNA-MS [37], iDNA-AB, and iDNA-ABT [38]. Table 4 lists the relevant information of the compared models.

**Table 4. Information on each comparison model.**

| Model | Description | References |
|---|---|---|
| iDNA-MS | KNFC, NCP and BPF three feature coding combined with random forest | [37] |
| iDNA-ABT | Transformers' bi-directional encoder architecture (BERT), using TIM loss | [38] |
| iDNA-AB | A variant of iDNA-ABT that differs by using the standard cross-entropy loss | |

All three are general-purpose methylation predictors that can predict various DNA methylation types, and all are DNA methylation prediction models proposed in the last three years. MEDCNN is compared with iDNA-MS in that the latter uses a conventional machine learning method to predict DNA methylation modification sites. While iDNA-AB and iDNA-ABT employ deep learning methods to construct their models, it is important to note that the features of DNA gene sequences are extracted without initially considering multiple dimensions. The comparison of this experiment will demonstrate the effectiveness of multidimensional information aided deep learning to predict DNA methylation sites. We used the same dataset in our validation and compared the prediction results of the above-mentioned models, and the resulting results are shown in S3 Table and Fig 7.

To more clearly compare the performance of the above-described models, we categorized the predictions into 100% to 90%, 90% to 80%, 80% to 70, and less than 70%, and tallied the distribution of predictions within the above-mentioned intervals for 17 datasets. The above-described are listed in Table 5.

ACC has been confirmed as the most intuitive measure of model performance. However, it has the obvious drawback that under the unbalanced negative and positive DNA methylation categories, the larger category will be the main factor for the ACC value. In contrast, MCC integrates the four values of TP, TN, FP, and FN, such that the model performance can be accurately indicated even under the unbalanced samples. Fig 8 presents the ACC and MCC values of the predicted results for 17 benchmark independent datasets.

To further examine whether there exists a distinction between the prediction outcomes of the compared models, a nonparametric Wilcoxon test was conducted independently for both the MEDCNN model and the other models. The Wilcoxon test was utilized to compare the discrepancies in ACC and MCC values between the two model. We calculated the rank sum statistic and the corresponding p-value to evaluate the dissimilarity between the two samples, with a significance level set at $\alpha = 0.05$. The calculation results are presented in Table 6.

Fig 7 presents the prediction results of the MEDCNN model on the datasets of several species; the mean values of SN and SP reached over the other three models, and the distribution of their results was also more above 80%, thus confirming the robust performance of the proposed model in predicting DNA methylation modification sites. As depicted in Fig 8A and Table 5, the MEDCNN model significantly outperformed the existing iDNA-MS model for 14 datasets in terms of ACC values, and the prediction results for the other three datasets were not significantly different, and six datasets achieved results above 90%. Besides, the MCC values in Fig 8B were significantly higher than iDNA-MS overall, such that the performance of using the deep learning method to predict DNA methylation modification sites was stronger compared with the conventional machine learning method. Similar results were presented compared with another deep learning model, iDNA-AB. The predicted ACC values of the MEDCNN model were 2.19% higher than those of iDNA-AB on average, with an increase of approximately 9% in 6mA_R. chinensis and 4.3% in 6mA_S. cerevisiae. As revealed by the above result, the feature matrix of DNA gene sequences extracted from multidimensional information can be more effective when methylation prediction is performed. The results of the Wilcoxon test in Table 6 indicated that the Wilcoxon signed-rank statistic for these groups

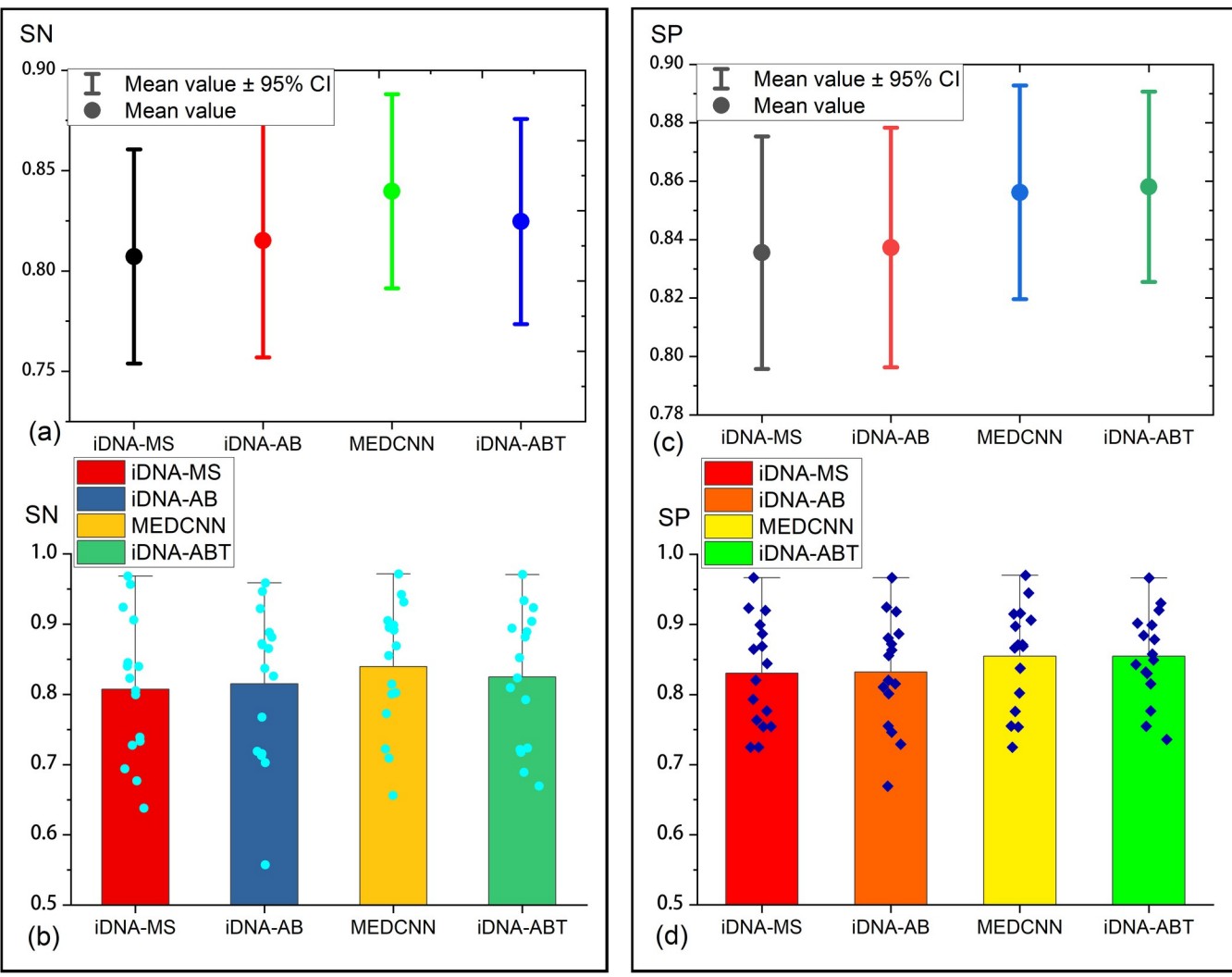

**Fig 7. Performance comparison between MEDCNN and other existing methods on 17 benchmark independent datasets.** (**a**) and (**b**) represent the comparison results of SN values for each model. (**c**) and (**d**) represent the comparison results of SP values for each model.

**Table 5. Statistics on the distribution of results for each comparison model in 17 datasets.**

| Evaluation index | Model | 100%-90% | 90%-80% | 80%-70% | 70%-0% |
|---|---|---|---|---|---|
| SN | iDNA-MS | 5 | 5 | 4 | 3 |
| | iDNA-AB | 4 | 7 | 5 | 1 |
| | MEDCNN | 5 | 8 | 3 | 1 |
| | iDNA-ABT | 5 | 6 | 4 | 2 |
| SP | iDNA-MS | 4 | 6 | 7 | 0 |
| | iDNA-AB | 4 | 9 | 3 | 1 |
| | MEDCNN | 5 | 8 | 4 | 0 |
| | iDNA-ABT | 5 | 9 | 3 | 0 |
| ACC | iDNA-MS | 3 | 8 | 5 | 1 |
| | iDNA-AB | 4 | 6 | 6 | 1 |
| | MEDCNN | 6 | 6 | 5 | 0 |
| | iDNA-ABT | 4 | 9 | 4 | 0 |

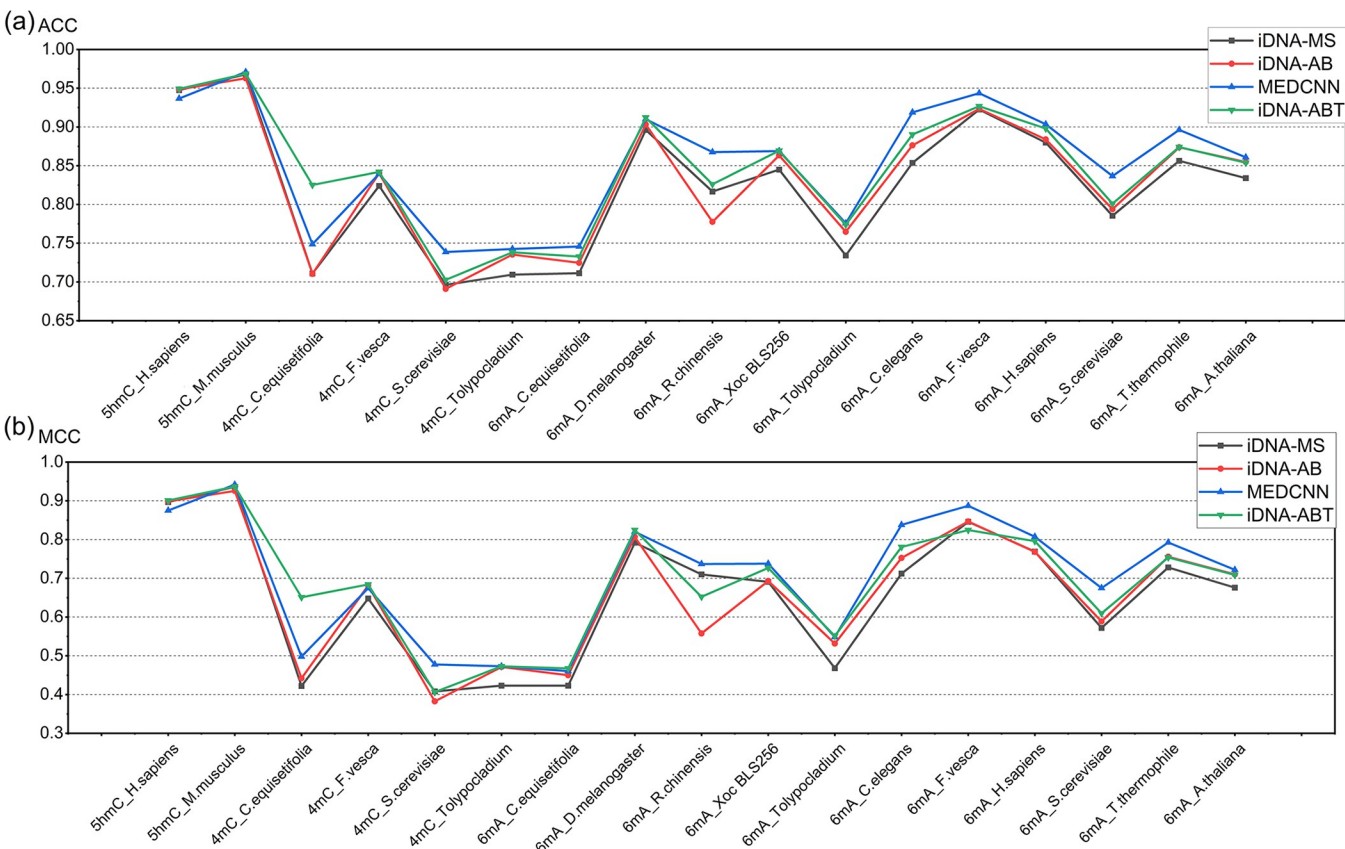

**Fig 8. Comparison of ACC and MCC values of the proposed MEDCNN and other existing models on 17 independent datasets.**

was smaller than the product of their sample sizes. Upon comparing the p-values, we discovered statistically significant differences between the MEDCNN model and other models in terms of all ACC values. This suggests that the MEDCNN model outperformed the above generic DNA methylation prediction models. The above-mentioned results suggest that the MEDCNN model can be more advantageous in predicting DNA methylation modification sites, with better prediction performance.

## Conclusions

A method was proposed in this study to extract features from multidimensional information of DNA gene sequences fused with deep learning to predict DNA methylation modification sites, which can predict multiple types of methylation (i.e., 4mC, 5hmC, and 6mA). The proposed method combines positional information of gene sequences, biological information and chemical information assisted convolutional neural network, such that DNA methylation

**Table 6. The results of the Wilcoxon test for each model.**

| Model group | ACC | | MCC | |
|---|---|---|---|---|
| | **Wilcoxon signed-rank statistic** | **p-value** | **Wilcoxon signed-rank statistic** | **p-value** |
| MEDCNN: iDNA-MS | 2 | 0.000046 | 45 | 0.145432 |
| MEDCNN: iDNA-AB | 9 | 0.000504 | 3 | 0.000076 |
| MEDCNN: iDNA-ABT | 34.5 | 0.044769 | 10 | 0.000656 |

modification sites can be predicted flexibly. By comparing with independent feature encoding methods and other advanced models for predicting DNA methylation sites, the experimental results indicated that the proposed method can achieve satisfactory results while increasing the accuracy of model prediction results. However, there is still room for improvement in the current study. Due to the lack of sufficient number of datasets of DNA methylation types for some species, the prediction results derived are not precise enough. Furthermore, the performance evaluation of the method again is worth refining when sufficient available datasets of DNA methylation are collected in the future.

## Supporting information

**S1 Table. The results of MEDCNN prediction for independent datasets 5hmC,4mC and 6mA.**
(DOCX)

**S2 Table. The results of different coding methods to predict the independent datasets 5hmC,4mC and 6mA.**
(DOCX)

**S3 Table. The results of different models predicted for independent datasets 5hmC,4mC and 6mA.**
(DOCX)

## Acknowledgments

We thank LetPub (www.letpub.com) for its linguistic assistance during the preparation of this manuscript.

## Author Contributions

**Conceptualization:** Mengshan Li.

**Data curation:** Lixin Guan.

**Formal analysis:** Lixin Guan.

**Funding acquisition:** Mengshan Li.

**Investigation:** Wenxing Hu.

**Methodology:** Wenxing Hu, Mengshan Li.

**Project administration:** Lixin Guan, Mengshan Li.

**Software:** Wenxing Hu.

**Supervision:** Lixin Guan, Mengshan Li.

**Validation:** Wenxing Hu.

**Visualization:** Wenxing Hu.

**Writing – original draft:** Wenxing Hu, Lixin Guan, Mengshan Li.

**Writing – review & editing:** Wenxing Hu, Lixin Guan, Mengshan Li.

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
