## [Decision Letter · Decision Letter 0]

29 Jun 2023

Dear Dr. Mengshan,

Thank you very much for submitting your manuscript "Prediction of DNA Methylation based on Multi-dimensional feature encoding and double convolutional fully connected convolutional neural network" for consideration at PLOS Computational Biology.

As with all papers reviewed by the journal, your manuscript was reviewed by members of the editorial board and by several independent reviewers. In light of the reviews (below this email), we would like to invite the resubmission of a significantly-revised version that takes into account the reviewers' comments.

We cannot make any decision about publication until we have seen the revised manuscript and your response to the reviewers' comments. Your revised manuscript is also likely to be sent to reviewers for further evaluation.

Sincerely,

Piero Fariselli

Academic Editor

PLOS Computational Biology

Sushmita Roy

Section Editor

PLOS Computational Biology

Reviewer's Responses to Questions

**Comments to the Authors:**

Reviewer #1: The authors present a novel machine learning-based model, MEDCNN, for the prediction of DNA methylation modification sites. While the paper is overall interesting, there are several aspects that require further clarification. Please find my specific comments below:

1. In the abstract, the authors mention that "most models have been built in terms of of a single methylation type. To address the above-mentioned issues, a deep learning-based method was proposed in this study for DNA methylation site prediction, termed the MEDCNN model.". However, based on the provided description, it seems that also the MEDCNN model has actually been trained separately on different species to predict a single methylation type. Please clarify this aspect for better understanding.

2. The dataset described in Table 1 appears to differ from the one used by Lv. et al. for training iDNA-MS. Notably, the dataset employed in this study includes a greater amount of training and testing data compared to Lv. et al. Moreover, it remains unclear how the training and testing data were generated. Could you please provide more details regarding the dataset acquisition process and the methodology used to partition the data into training and testing sets? Specifically, does the dataset exhibit any sequence similarity between the training and testing sequences?

3. The limited description of the training/testing data raises concerns regarding the comparison results presented in the section titled "Experimental results compared with other models." Several points deserve clarification:

- Is it possible that the prediction performance of MEDCNN is influenced by the larger training dataset used?

- To ensure a robust comparison with iDNA-MS, it would be better to use precisely its training and testing data. For instance, could there be an overlap between the test set used here and training set of iDNA-MS?

- Overall, figures 6 and 7 appear to exhibit comparable results, without any discernible predictor outperforming the other. It is advisable to conduct statistical significance tests, such as t-tests or Wilcoxon tests, to determine whether there are statistically significant differences in prediction performance.

4. I have some concerns regarding the effectiveness of the NCP features utilized in the model. Do these features genuinely contribute to improved performance? They seem akin, albeit not formally, to a one-hot encoding of the four nucleotides. It is reasonable to assume that a machine learning approach would readily learn a mapping between BFP and NCP, as it represents a one-to-one mapping between binary vectors. Additionally, there appears to be little discrepancy in performance between these two features when used individually, as demonstrated in the "Experimental results of different feature encoding" section. Kindly provide further justification for the necessity of these features. Also in this case, employing a statistical significance test could aid in elucidating whether certain features or their combinations outperform others.

5. Regarding the section titled "Experimental results of cross-species validation," the concluding statement appears unsupported by the experimental tests depicted in Figure 5 since the authors' assertion appears valid solely for C. equisetifolia.

6. The manuscript would benefit from a thorough proofreading to rectify typos. I have compiled a list of some identified errors:

- Page 9, Equation 6: The term "D_i" lacks an explanation. Also, it has been possibly mistyped as "D_I".

- Page 9, line 182: The sentence beginning with "However, since a 2D feature ..." requires revision.

- Page 12, line 254: The meaning of "to identify 17 datasets" is unclear.

- Page 12, Figure 4(a): The labels BFP, NCP, and DPCP are probably a typo. If not, please clarify their meaning.

- Page 13, line 176: Add a period before "Five datasets."

- Page 13, line 279: Please revise the sentence starting with "predict whether the weather."

- Page 15, line 304: Please revise the sentence beginning with "iDNA-AB and iDNA-ABT, ..."

Reviewer #2: The authors proposed a deep learning-based method for DNA methylation site prediction, termed the MEDCNN model. The MEDCNN model extracted feature information from gene sequences in three dimensions (i.e., positional information, biological information, and chemical information). The proposed method employs a convolutional neural network model with double convolutional layers and double fully connected layers while iteratively updating the gradient descent algorithm. Besides, the MEDCNN model can predict different types of DNA methylation. The deep learning method based on coding from multiple dimensions outperformed single coding methods.

There are some major concerns:

1) In Fig3 abcde, the authors illustrated the profiles or distributions for each measure, but the readers cannot understand what they mean by the profiles. Explain why the figures have profiles clearly. They should explain details in the figures intelligibly in the figure legends.

2) The authors employed convolutional block attention module after the convolutional layer to gain insights into the important features and their locations along the channels and spatial axes by reasoning about their attention images. However, they did not show the attention map and did not explain the insights on the important features. Does the addition of the attention module contribute to the improvement in prediction performance?

3) It may be hard to understand that MDECNN outperformed the existing methods, because the authors did not investigate the latest methods for the respective methylation site prediction in DNA.

4) https://github.com/gnnumsli/DNA-Methylation is immature. This URL site must be prepared so that the users can execute the program and obtain the same results as provided by the manuscript.

**Have the authors made all data and (if applicable) computational code underlying the findings in their manuscript fully available?**

Reviewer #1: Yes

Reviewer #2: **No: **Thier URL site must be prepared so that the users can execute the program and obtain the same results as provided by the manuscript.

PLOS authors have the option to publish the peer review history of their article (what does this mean?). If published, this will include your full peer review and any attached files.

Reviewer #1: No

Reviewer #2: No
---

## [Decision Letter · Decision Letter 1]

18 Jul 2023

Dear Dr. Li,

We are pleased to inform you that your manuscript 'Prediction of DNA Methylation based on Multi-dimensional feature encoding and double convolutional fully connected convolutional neural network' has been provisionally accepted for publication in PLOS Computational Biology.

Best regards,

Piero Fariselli

Academic Editor

PLOS Computational Biology

Sushmita Roy

Section Editor

PLOS Computational Biology

Reviewer's Responses to Questions

**Comments to the Authors:**

Reviewer #1: All my previous concerns have been addressed by the authors in the revised version of the manuscript. I have no further remarks this time

Reviewer #2: It is improved according to my comments

**Have the authors made all data and (if applicable) computational code underlying the findings in their manuscript fully available?**

Reviewer #1: Yes

Reviewer #2: Yes

PLOS authors have the option to publish the peer review history of their article (what does this mean?). If published, this will include your full peer review and any attached files.

Reviewer #1: No

Reviewer #2: No

---

## [Editor Report · Acceptance letter]

14 Aug 2023

PCOMPBIOL-D-23-00775R1 

Prediction of DNA Methylation based on Multi-dimensional feature encoding and double convolutional fully connected convolutional neural network

Dear Dr Li,

I am pleased to inform you that your manuscript has been formally accepted for publication in PLOS Computational Biology. Your manuscript is now with our production department and you will be notified of the publication date in due course.

With kind regards,

Zsuzsanna Gémesi
